# In-Hospital Mortality and Risk Prediction in Minimally Invasive Sutureless versus Conventional Aortic Valve Replacement

**DOI:** 10.3390/jcm11247273

**Published:** 2022-12-07

**Authors:** Giuseppe Santarpino, Roberto Lorusso, Armin Darius Peivandi, Francesco Atzeni, Maria Avolio, Angelo Maria Dell’Aquila, Giuseppe Speziale

**Affiliations:** 1Department of Experimental and Clinical Medicine, Magna Graecia University, 88100 Catanzaro, Italy; 2Department of Cardiac Surgery, Città di Lecce Hospital, GVM Care & Research, 73100 Lecce, Italy; 3Department of Cardiac Surgery, Paracelsus Medical University, 90471 Nuremberg, Germany; 4Cardio-Thoracic Surgery Department, Heart & Vascular Centre, Maastricht University Medical Hospital, 6229 HX Maastricht, The Netherlands; 5Department of Cardiac Surgery, Münster Universität, 48143 Münster, Germany; 6Clinical Data Management, GVM Care & Research, 00137 Rome, Italy; 7Department of Cardiac Surgery, Anthea Hospital, GVM Care & Research, 70124 Bari, Italy

**Keywords:** aortic stenosis, aortic valve replacement, minimally invasive surgery, sutureless valves

## Abstract

Objective. Available evidence suggests that a minimally invasive approach with the use of sutureless bioprostheses has a favorable impact on the outcome of patients undergoing aortic valve replacement (AVR). Methods. From 2010 to 2019, 2732 patients underwent conventional AVR through median sternotomy with a stented bioprosthesis (*n* = 2048) or minimally invasive AVR with a sutureless bioprosthesis (*n* = 684). Results. Using the propensity score, 206 patients in each group were matched, and the matched groups were well balanced regarding preoperative risk factors. Both unmatched and matched patients of the sutureless + minimally invasive group showed significantly shorter cross-clamp times and longer ICU stay. In-hospital mortality was the only outcome measure that was confirmed in both analyses, and was higher in the stented + conventional group (2.54% and 2.43% in unmatched and matched patients, respectively) compared with the sutureless + minimally invasive group (0.88% and 0.97% in unmatched and matched patients, respectively) (*p* = 0.0047 and *p* < 0.0001, respectively). No differences in postoperative pacemaker implantation were recorded in matched patients of both groups (*n* = 2 [1%] in the stented + conventional group vs. *n* = 4 [2%] in the sutureless + minimally invasive group; *p* = 0.41). The discrimination power of EuroSCORE II was not confirmed in the sutureless + minimally invasive group, yielding an area under the ROC curve of 0.568. Conclusions. Minimally invasive sutureless AVR has a favorable impact on the immediate outcome and is associated with significantly lower in-hospital mortality rates compared with conventional AVR, resulting in the absence of the discrimination power of EuroSCORE II for predicting AVR outcomes.

## 1. Introduction

Over the last several years, the optimal treatment option for aortic valve stenosis has been a subject of intense debate. The guideline indications for surgery have changed since the introduction of transcatheter aortic valve implantation (TAVI), and major interest in this issue remains given the increasing prevalence of aortic stenosis with advancing age [1].

The use of new prosthetic models and the adoption of minimally invasive approaches were initially demonstrated to be safe in patients at high or prohibitive surgical risk and have then gradually moved to lower risk patients. The new frontier of research has therefore concentrated on low-risk patients undergoing surgical aortic valve replacement (SAVR), for whom current ESC/EACTS guidelines are somewhat “unclear”: SAVR is recommended in low-risk patients (STS-PROM/EuroSCORE II <4%) or unsuitable for TAVI and operable [1]. Then, should a low-risk patient who is both operable and suitable for TAVI undergo a transcatheter procedure? Indeed, this kind of patient falls into the so-called “remaining patients” category [1], who may undergo both procedures.

Interestingly, no mentioning has been made in the current guidelines about the use of sutureless and rapid-deployment prostheses that may reduce cross-clamp and cardiopulmonary bypass times, and potentially lower perioperative complications of SAVR. As stated in the guidelines, the lack of large-scale randomized trials in this context—though a published trial does exist [2]—makes SAVR with conventional prosthetic valves the gold standard.

Similarly, minimally invasive surgery is not mentioned in the guidelines, but it is acknowledged that SAVR via full sternotomy may contribute to the development of pulmonary complications [1,3,4]. However, available evidence suggests that a minimally invasive approach has a favorable impact on the immediate outcome and is associated with lower mortality rates compared to standard sternotomy [5].

The aim of this study was to assess if minimally invasive surgery with a sutureless valve may result in a better outcome compared with full sternotomy with a stented bioprosthesis in low-risk patients and in the “remaining patients” category so as to support the Heart Team decision-making tailored to the individual patient.

## 2. Methods

From 2010 to 2019, data of all patients referred to nine cardiac surgery centers of the GVM Care and Research Group (Anthea Hospital, Bari, Italy; Città di Lecce Hospital, Lecce, Italy; ICLAS, Rapallo, Italy; Maria Cecilia Hospital, Cotignola, Italy; Maria Eleonora Hospital, Palermo, Italy; Maria Pia Hospital, Turin, Italy; Salus Hospital, Reggio Emilia, Italy; Santa Maria Hospital, Bari, Italy; Villa Torri Hospital, Bologna, Italy), with symptomatic severe aortic stenosis or with either steno-insufficiency with an indication for surgery after evaluation by the Heart Team, were retrieved from a single, centralized electronic data management system.

All patients aged >60 years who had undergone surgical bioprosthetic aortic valve replacement were included in this analysis.

All patients underwent conventional aortic valve replacement (AVR) through longitudinal median sternotomy or minimally invasive AVR via a ministernotomy or a right anterior minithoracotomy, according to the surgeon’s experience and preference. Similarly, if a bioprosthetic aortic valve was used (usually in patients >65 years old), the choice to implant a stented or stentless valve, or a sutureless bioprosthesis, was left to the surgeon at the time of operation. However, sutureless valves were less frequently implanted in patients undergoing conventional AVR as we selected patients that were operated by experienced surgeons that had completed the learning curve.

Patients were divided into two groups based on the type of prosthetic valve used and the surgical approach they received: the stented + conventional group (*n* = 2048) undergoing conventional AVR with a stented bioprosthesis (the stented prostheses were either the Mosaic Ultra or the Avalus, both by Medtronic, MN, USA), and the sutureless + minimally invasive group (n = 684) undergoing minimally invasive AVR with the Perceval bioprosthesis (Corcym, Milan, Italy). A propensity score matching (PSM) analysis was used to address potential selection bias from a lack of randomization. For the matched pair samples, postoperative clinical data and hospital costs were obtained.

The study was approved by a human research ethical review board (IRB 2/2021, 19 October 2021).

The primary outcome measures were in-hospital mortality, hospital costs, cardiopulmonary bypass (CPB) and cross-clamp times, intensive care unit (ICU) and hospital stay, need for blood transfusion, and postoperative pacemaker implantation.

### 2.1. Surgical Approach

In the minimally invasive group, a partial J-shaped ministernotomy in the third to fourth intercostal space or a right anterior thoracotomy in the second intercostal space was performed. For both surgical approaches, CPB was established with central arterial and central or peripheral venous cannulation. Antegrade crystalloid cardioplegia was used. The stented prostheses were implanted with semi-continuous sutures or U-stitches with pledgets according to the surgeon’s preference.

The implant technique of the Perceval valve has been described previously [6], along with the tips and tricks to minimize the risk for postoperative pacemaker implantation [7], which have been adopted by all surgeons involved in the study after appropriate training provided by the GVM Care and Research Group.

### 2.2. Statistical Analysis

Statistics were performed using MedCalc Software (MecCalc Software Ltd., Ostend, Belgium). Normality of continuous variables was tested by the Shapiro–Wilk test. Continuous variables are depicted as the median and interquartile range. Categorical variables are reported as counts and percentages. To provide a balanced data frame of patients with the same likelihood of undergoing minimally invasive sutureless AVR or conventional stented AVR, a PSM was performed according to the following: prior to matching, the influence of preoperative values (Table 1) on the decision of minimally invasive sutureless AVR was assessed by univariate logistic regression. Significant values, except EuroSCORE II (as it is a composite of preoperative values), were included in a multivariate logistic regression model for developing a propensity score. Variables not included in the model by statistical software are shown in Table 1. Statistical significance was set at *p* ≤ 0.05. The propensity score was defined as the probability of receiving minimally-invasive sutureless valve replacement. After creation of the propensity score, a two-decimal digit case-control matching based on the propensity score was performed. In the matched cohort, univariate logistic regression analysis revealed no significant differences in preoperative parameters between groups as a sign of good matching (Table 1). For comparison of results in the unmatched cohort, unpaired testing was applied: continuous non-normally distributed variables were compared using a Mann–Whitney U test, and for dichotomous variables, the Chi-square test was performed. After matching, paired testing was applied as suggested by Bland and Altman [8]: continuous non-normally distributed variables were compared using the Wilcoxon test. The McNemar test was applied to dichotomous variables.

Validation analysis of EuroSCORE II was conducted as described in the ABCD model by Steyerberg and Vergouwe [9]. Calibration was assessed by slope and intercept analysis. Receiver operating characteristic (ROC) curve analysis was conducted for discrimination between groups (Figure 1).

## 3. Results

The preoperative characteristics of the study population are reported in Table 1. Unmatched patients of the stented + conventional group were older and at higher surgical risk compared with patients of the sutureless + minimally invasive group. The latter showed clinically pure aortic stenosis more often. The cases of concomitant surgery are those in which the aortic valve replacement has been associated with a septal myectomy.

Using the propensity score, 206 patients in each group were matched, and the matched groups were well balanced regarding preoperative risk factors (Table 1).

Postoperative results significantly differed between groups (Table 2). The matched stented + conventional group showed prolonged CPB and cross-clamp times and longer hospital stay than the sutureless + minimally invasive group, but differences were no longer present after PSM (Table 2). In contrast, both unmatched and matched patients of the sutureless + minimally invasive group showed significantly shorter cross-clamp times and longer ICU stay. Hospital costs and the need for blood transfusion were higher in unmatched stented + conventional patients, but the opposite was seen in matched patients of the same group.

In-hospital mortality was the only outcome measure that was confirmed in both analyses, and was higher in the stented + conventional group (2.54% and 2.43% in unmatched and matched patients, respectively) compared with the sutureless + minimally invasive group (0.88% and 0.97% in unmatched and matched patients, respectively) (*p* = 0.0047 and *p* < 0.0001, respectively).

No differences in postoperative pacemaker implantation were recorded in matched patients of the two groups (*n* = 2 [1%] in the stented + conventional group vs. *n* = 4 [2%] in the sutureless + minimally invasive group; *p* = 0.41).

Postoperatively, in the matched population, three cerebrovascular events (1.4%; two transient ischemic attacks and one permanent neurologic deficit) were recorded in the stented + conventional group vs. one event (0.5%; one transient ischemic attack) in the sutureless + minimally invasive group (*p* = 0.31).

In the whole study population, the area under the ROC curve for EuroSCORE II was 0.696 (Figure 1A) indicating a good discrimination power. The same applies to the stented + conventional group showing an area under the ROC curve of 0.7 (Figure 1B). On the contrary, the discrimination power of EuroSCORE II was not confirmed in the sutureless + minimally invasive group, yielding an area under the ROC curve of 0.568 (Figure 1C).

## 4. Discussion

Our results show that minimally invasive AVR with a sutureless bioprosthesis in patients with aortic valve stenosis is associated with significantly shorter ischemic times and lower mortality rates compared with matched patients undergoing conventional SAVR, resulting in the absence of the discrimination power of EuroSCORE II for predicting AVR outcomes. Although our study population included patients at low and intermediate surgical risk with a median EuroSCORE II of 2.23 in the selected matched cohort, this finding should be part of the Heart Team decision-making when evaluating SAVR vs. TAVI.

Even if no randomized trials have been conducted as yet on minimally invasive AVR with sutureless bioprostheses, it is difficult to understand why none of the two approaches, either in isolation or combined, have not been addressed in the recent ESC/EACTS guidelines [1].

In the prospective randomized PERSIST-AVR trial [2], sutureless valves significantly reduced surgical times and were non-inferior to stented valves with respect to major adverse cerebral and cardiovascular events at 1 year, suggesting that sutureless valves should be considered as part of a comprehensive valve program. However, patients undergoing AVR through a minithoracotomy were excluded from this study, making the assessment of the potential benefit of minimally invasive surgery with a sutureless bioprosthesis impossible.

Similarly to our study, Dalén et al. [10] analyzed early postoperative outcomes after AVR through a ministernotomy with a sutureless bioprosthesis compared with a full sternotomy with a stented bioprosthesis, showing that the former was associated with shorter CPB and aortic cross-clamp times than the latter. This is noteworthy given that the minimally invasive approach is generally considered more demanding and time-costing than open heart surgery. Additionally, patients undergoing ministernotomy received less packed red blood cells, but the technique used at that time was associated with a higher risk for postoperative pacemaker implantation. However, more recently, rates of postoperative pacemaker implantation after sutureless AVR have dramatically declined after the learning curve has been overcome [11,12]. In our study, although we recorded a twofold higher rate of postoperative pacemaker implantation in the matched sutureless + minimally invasive group compared with the stented + conventional group, it accounted for a very low percentage of patients (2%) and did not significantly differ between groups.

In addition, Pollari et al. [13] compared patients undergoing AVR with a sutureless valve vs. a stented valve showing a better short-term outcome in the sutureless group after PSM, with a total hospital cost saving of approximately 25%. However, authors’ conclusions were only derived from the faster procedural time of sutureless AVR. Our multicenter study, by evaluating the effect of using a minimally invasive approach in patients referred to a variety of centers with different experience levels, allows for drawing considerations on the reproducibility of the results and on several issues related to different management protocols across the participating centers. However, despite the differences in the strategies adopted in the various centers (e.g., indications for triggering blood transfusion, length of ICU stay), mortality rates remained significantly lower in the matched sutureless + minimally invasive group. Moreover, as also intuitively expected given that patients in this group were at a lower surgical risk than unmatched patients receiving a stented valve, the EuroSCORE II lost its predictive ability also in the overall group.

One of the limitations of our article is that the patients who underwent a minimally invasive approach were operated on by a group of surgeons with a more advanced learning curve than the patients operated on with a conventional approach. However, it should be emphasized that, contrary to popular belief, the full sternotomy is still the “standard of care” in case of aortic valve replacement. Consequently, also considering the high number of surgeons involved in our study with different levels of experience, our study reflects a real-life setting.

We also want to underline the originality of our study that, unlike other previous studies [10,14], which compared minimally invasive rapid-deployment or sutureless prostheses versus conventional approaches, we here recorded a significant difference in hospital mortality. Therefore, given the debated results, the need for a “truly” randomized trial is mandatory.

To the best of our knowledge, this is the first study demonstrating a significant impact of minimally invasive AVR using sutureless bioprostheses on in-hospital mortality. This finding was based on the data recorded by nine cardiac surgery centers and cannot be affected by the different protocols in use but rather reflects a real-life scenario.

In a prior study from our group that assessed the potential advantages of using sutureless vs. conventional prostheses for minimally invasive AVR with data collected from the same centralized electronic data management system, similar favorable outcomes were reported with a 30-day mortality of 0.7% and 2.1% in patients receiving a sutureless and a conventional prosthesis, respectively (*p* = 0.076) [15]. However, a minimally invasive strategy was used in both patient groups in this study.

In intermediate or high-risk patients, SAVR is associated with longer lasting results, and a more favorable cost-effectiveness ratio compared with TAVI is mostly attributable to the higher cost of transcatheter devices [16]. In our analysis involving low-risk patients, no differences in healthcare costs were observed between unmatched groups despite the higher cost of sutureless devices compared to conventional devices. In contrast, healthcare costs were higher in the matched sutureless group, and it would be interesting to know if costs varied according to the protocols used for blood transfusion, length of ICU, and hospital stay, etc., but this is a limitation of our study. However, either similar or higher healthcare costs are associated with a significantly lower in-hospital mortality in the sutureless + minimally invasive group. The question is whether an average cost of additional EUR 700.00 may be worth it to achieve a significant reduction in mortality in this patient subset.

Moreover, our suggested approach of minimally invasive sutureless AVR, though more expensive, is more effective and matches well with the incremental cost-effectiveness ratio calculation, resulting in a cost of EUR 479.45 per additional in-hospital life saved. However, the question of whether the gain in reduced in-hospital mortality is worth the cost remains open.

EuroSCORE II has a strong predictive ability that has recently been confirmed using data collected from our centralized database, and performs better than a parsimonious risk score [17]. In our study, in patients treated with a minimally invasive approach with a sutureless valve, the observed risk was much lower than predicted. This suggests a protective effect conferred by our strategy that should always be evaluated during preoperative planning and adopted in anatomically suitable patients (i.e., without type 0 bicuspid aortic valve).

Despite similar clinical outcomes across the different participating centers, the length of ICU stay ranged from 0.9 to 2.5 days and the length of hospital stay ranged from 12 to 16 days, where a shorter ICU stay was usually followed by a longer hospital stay. The effect of the management protocols on the length of ICU and hospital stay—which may also be observed for blood transfusion trigger/cut-off—is a clear bias and a limitation of our study, given that the centers where a longer ICU stay was recorded were those where sutureless prostheses were most often used.

## 5. Conclusions

In conclusion, despite the inherent limitations of our multicenter, an observational, real-life study, partially addressed by using PSM, minimally invasive sutureless AVR was associated with significantly lower in-hospital mortality rates compared with conventional surgery, and this treatment option should be considered in patients with favorable anatomical characteristics. Further evaluation in a randomized trial combining these two procedural aspects is urgently warranted as no indications are provided in the current guidelines and information provided by independent studies is constantly growing.

## Figures and Tables

**Figure 1 jcm-11-07273-f001:**
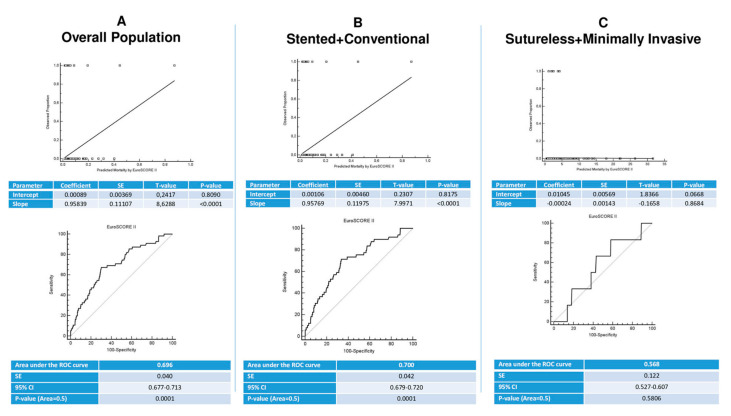
Calibration and discrimination of EuroSCORE II in the overall population. CI, confidence interval; SE, standard error; ROC, receiver operating characteristic.

**Table 1 jcm-11-07273-t001:** Preoperative characteristics of the study population before and after propensity score matching.

	**Unmatched**							
	**Stented + Conventional (*n* = 2048)**			**Sutureless + Minimally Invasive (*n* = 684)**			***p*-Value** **(uni log reg)**	***p*-Value** **(multi log reg)**
	**Median/N**	**1st Percentile**	**3rd Percentile**	**Median/N**	**1st Percentile**	**3rd Percentile**		
Male sex	981	47.90	%	250	36.55	%	0.00001	0.0233
Age, years	77	72	81	78	73	82	0.00001	0.0008
Emergency	226	11.04	%	53	7.75	%	0.0145	Not included in the model
Active endocarditis	33	1.61	%	1	0.15	%	0.0196	Not included in the model
Previous endocarditis	19	0.93	%	0	0	%	0.9844	
Creatinine preop, mg/dL	0.9	0.8	1.1	0.9	0.7	1.1	0.1492	Not included in the model
COPD	122	5.96	%	67	9.80	%	0.0002	Not included in the model
PAP >30 mmHg	627	30.62	%	411	60.09	%	0.00001	0.0315
History of syncope	198	9.67	%	13	1.90	%	0.0138	0.0806
EuroSCORE II, %	2.8	1.74	4.7	2.14	1.39	3.44	0.00001	
LVEF preop, %	55	50	60	60	55	60	0.00001	0.0005
NYHA class III or IV	1121	54.74	%	187	27.34	%	0.00001	Not included in the model
Isolated aortic valve stenosis	1671	81.59	%	588	85.96	%	0.0091	Not included in the model
	**Matched**							
	**Stented + Conventional (*n* = 206)**			**Sutureless + Minimally Invasive (*n* = 206)**			***p*-Value**	
	**Median/N**	**1st Percentile**	**3rd Percentile**	**Median/N**	**1st Percentile**	**3rd Percentile**		
Male sex	77	37.38	%	67	32.52	%	0.3018	
Age, years	79	75	83	78	74	82	0.2611	
Emergency	14	6.80	%	12	5.83	%	0.6856	
Active endocarditis	4	1.94	%	0	0.00	%	0.98	
Previous endocarditis	2	0.97	%	0	0.00	%	0.9859	
Creatinine preop., mg/dL	0.9	0.7	1.1	0.8	0.7	1	0.2778	
COPD	15	7.28	%	18	8.74	%	0.5866	
PAP >30 mmHg	74	35.92	%	71	34.47	%	0.757	
History of syncope	12	5.83	%	13	6.31	%	0.8366	
EuroSCORE II, %	2.84	1.71	4.495	2.23	1.47	3.7	0.0928	
LVEF preop., %	55.5	55	60	60	55	60	0.9437	
NYHA class III or IV	108	52.43	%	126	61.17	%	0.0738	
Isolated aortic valve stenosis	171	83.01	%	173	83.98	%	0.7907	

COPD, chronic obstructive pulmonary disease; LVEF, left ventricular ejection fraction; NYHA, New York Heart Association; PAP, pulmonary artery pressure; preop, preoperatively.

**Table 2 jcm-11-07273-t002:** Postoperative results before and after propensity score matching.

	**Unmatched**						
	**Stented + Conventional (*n* = 2048)**			**Sutureless + Minimally Invasive (*n* = 684)**			***p*-Value**
	**Median/N**	**1st Percentile**	**3rd Percentile**	**Median/N**	**1st Percentile**	**3rd Percentile**	
In-hospital mortality, %	52	2.54	%	6	0.88	%	0.047
Hospital costs, €	24,181.5	20,486.6	24,675	20,896.33	20,486.6	24,675	0.4594
CPB time, min	73	58.75	88	56	43	71	<0.0001
Cross-clamp time, min	57.5	45	69	42	34	53	<0.0001
ICU stay, days	1.77	0.95	2	1.92	1.59	2.59	<0.0001
Hospital stay, days	11	8	15	10	8	13	<0.0001
Transfusions	664	32.42	%	146	21.35	%	0.065
	**Matched**						
	**Stented + Conventional (*n* = 206)**			**Sutureless + Minimally Invasive (*n* = 206)**			***p*-Value**
	**Median/N**	**1st Percentile**	**3rd Percentile**	**Median/N**	**1st Percentile**	**3rd Percentile**	
In-hospital mortality, %	5	2.43	%	2	0.97	%	<0.0001
Hospital costs, €	23,479.46	20,486.6	24,675	24,181.5	20,486.6	24,675.19	0.0118
CPB time, min	73	60	91.5	65	52.75	78.25	0.0627
Cross-clamp time, min	58	45	70.5	48	40	60	0.0139
ICU stay, days	1.65	0.92	1.96	1.92	1.74	2.56	<0.0001
Hospital stay, days	11	8	14	11	9	14	0.7835
Transfusions	65	31.55	%	81	39.32	%	0.0001

CBP, cardiopulmonary bypass; ICU, intensive care unit.

## Data Availability

Data is available by request to the corresponding author.

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
