# Peer review of "In-Hospital Mortality and Risk Prediction in Minimally Invasive Sutureless versus Conventional Aortic Valve Replacement"

_jcm, 2022, doi:10.3390/jcm11247273_

Round 1

Reviewer 1 Report

Santarpino and colleagues show in their manuscript the better outcomes of sutureless and minimally invasive approach in patients with severe aortic stenoss. The approach with propensity matching seems sound from the perspective of the reviewer. One minor element would be the disclosure which types of heart valves were used in the patients and if the patients had any concomitant surgery like coronary bypass grafting? After adding these additional information the manuscript is quite good.

Author Response

We thank the reviewer for the comments on our paper. We have added data on the prosthetic models and the concomitant procedures as required.

Reviewer 2 Report

Thank you for inviting me to review the article entitled “In-hospital mortality and risk prediction in minimally invasive sutureless versus conventional aortic valve replacement” submitted by Santarpino and colleagues.

The authors aimed to analyze the impact of minimally invasive AVR with a sutureless bioprosthesis on the in-hospital mortality. A total of 206 patients were matched by preoperative characteristics included in the euroSCORE using the propensity score with another 206 patients receiving a standard sutured aortic bioprosthesis through full sternotomy. In-hospital mortality was higher in the standard group (p<0.001). The authors concluded that MI-AVR has a favorable impact on early mortality and the discrimination power of the euroSCORE II was not confirmed in this patient cohort (ROC of 0.568).

One major limitation of the study is that, as described in the methods, the patients operated through a minimally invasive approach were selected patients operated by experienced surgeons. Isolated AVR through minimally invasive approaches, mainly through UHS is becoming otherwise more and more a standard of care. However this patient population is not reflecting the real-life setting.

The novelty of this research is poor. To name other studies, Borger et al showed in a randomized multicentric study (CADENCE-MIS), comparing patients receiving a rapid-deployment AVR through a minimally invasive approach with conventional AVR through full sternotomy no difference in mortality or other early events. Similarly to this study, the cross-clamp times were significantly reduced in the rapid-deployment minimally invasive cohort (Michael A. Borger, Vadim Moustafine, Lenard Conradi, Christoph Knosalla, Markus Richter, Denis R. Merk, Torsten Doenst, Robert Hammerschmidt, Hendrik Treede, Pascal Dohmen, Justus T. Strauch, A Randomized Multicenter Trial of Minimally Invasive Rapid Deployment Versus Conventional Full Sternotomy Aortic Valve Replacement, The Annals of Thoracic Surgery,Volume 99, Issue 1, 2015, Pages 17-25, ISSN 0003-4975, https://doi.org/10.1016/j.athoracsur.2014.09.022.) Magnus Dalén and colleagues  also showed in a PMS analysis no difference in early mortality  between sutureless minimally invasive AVR and sutured full sternotomy in 171 paired patients. (Aortic valve replacement through full sternotomy with a stented bioprosthesis versus minimally invasive sternotomy with a sutureless bioprosthesisEuropean Journal of Cardio-Thoracic Surgery, Volume 49, Issue 1, January 2016, Pages 220–227, https://doi.org/10.1093/ejcts/ezv014).

Author Response

We thank the reviewer for his comments, we have now highlighted the difference in the learning curve between the various surgeons in the paper. We also added the suggested articles in the references (Dalen et al was already present), and highlighted the difference and originality of our paper compared to the other articles, as well as underlined our limitations.

Reviewer 3 Report

Santarpino and colleagues compared clinical outcomes in 2732 patients underwent conventional AVR through median sternotomy with a stented bioprosthesis (n=2048) and minimally invasive AVR with a sutureless bioprosthesis (n=684). They concluded that minimally invasive sutureless AVR was associated with significantly lower in-hospital mortality rates compared with conventional surgery and this treatment option should be considered in patients with favorable anatomical characteristics. This study is very well written in English. Statistical analysis is precise and accurate. It comes to very interesting conclusions, especially in this era in which the treatment of aortic valvular disease has important technical evolutions. Congratulations to the authors.

Author Response

Thank you very much for your comments!

Round 2

Reviewer 1 Report

The authors answered to all comments.

Reviewer 2 Report

Thank you for the corrections and small improvements.